# Prevalence of Myocardial Injury and Myocardial Infarction in Patients with a Hypertensive Emergency: A Systematic Review

**DOI:** 10.3390/diagnostics13010060

**Published:** 2022-12-26

**Authors:** Mohammed A. Talle, Ellen Ngarande, Anton F. Doubell, Philip G. Herbst

**Affiliations:** 1Division of Cardiology, Department of Medicine, Faculty of Medicine and Health Sciences, Stellenbosch University and Tygerberg Academic Hospital, Cape Town 7505, South Africa; 2Department of Medicine, Faculty of Clinical Sciences, College of Medical Sciences, University of Maiduguri and University of Maiduguri Teaching Hospital, Maiduguri 600004, Nigeria

**Keywords:** hypertensive emergency, myocardial infarction, myocardial infarction subtypes, myocardial injury

## Abstract

Myocardial injury and myocardial infarction can complicate a hypertensive emergency, and both are associated with poor prognosis. However, little is known about the prevalence of myocardial injury and the different subtypes of myocardial infarction in patients with hypertensive emergencies. This systematic review aims to determine the prevalence of myocardial infarction and its subtypes, and the prevalence of myocardial injury in patients with hypertensive emergencies following the PRISMA guideline. A systematic search of PubMed, Web of Science, and EBSCOHost (MEDLINE) databases was carried out from inception to identify relevant articles. A total of 18 studies involving 7545 patients with a hypertensive emergency were included. Fifteen (83.3%) studies reported on the prevalence of myocardial infarction ranging from 3.6% to 59.6%, but only two studies specifically indicated the prevalence of ST-elevation and non-ST-elevation myocardial infarction. The prevalence of myocardial injury was obtained in three studies (16.7%) and ranged from 15% to 63%. Despite being common, very few studies reported myocardial injury and the subtypes of myocardial infarction among patients presenting with a hypertensive emergency, highlighting the need for more research in this area which will provide pertinent data to guide patient management and identify those at increased risk of major adverse cardiovascular events.

## 1. Introduction

Despite advances in our knowledge and understanding of systemic hypertension, the prevalence and complications associated with hypertension remain high. Since 1990, the number of people living with systemic hypertension has doubled across the world, with low- and middle-income countries accounting for 75% of the global burden of hypertension [1]. Systemic hypertension is the most prevalent cardiovascular risk factor resulting in premature death and accounts for more than 50% of cases of stroke, myocardial infarction (MI), and heart failure, globally [2]. Hypertension could remain asymptomatic and be incidentally diagnosed, present with features of chronic target organ damage due to long-standing uncontrolled blood pressure, or present as a hypertensive emergency. A hypertensive emergency results from an acute severe rise in blood pressure and is characterized by acute hypertension-mediated organ damage involving the heart, brain, kidneys, retina, and aorta. This mostly occurs in persons with pre-existing hypertension but could occur de novo without a history of pre-existing hypertension or its treatment. Although the availability of safe, efficient, and well-tolerated anti-hypertensive medication has resulted in a significant reduction in mortality in patients with a hypertensive emergency, the prevalence remains unchanged at about 0.2% [3,4,5,6]. The different acute hypertension-mediated organ damages share a common pathophysiologic mechanism that involves the disruption of vascular autoregulation, vascular endothelial dysfunction, and widespread activation of the renin angiotensin aldosterone system across vascular beds.

Acute hypertension-mediated organ damage commonly involves the cardiovascular system and presents as an acute coronary syndrome (ACS)/MI, acute heart failure/cardiogenic pulmonary edema, and rarely, acute aortic syndrome. The most common cardiac manifestation is an acute pulmonary edema/acute heart failure and ACS [7]. In general, the prognosis is poorer in patients with hypertension complicated by acute hypertension-mediated organ damage than without [8], and the most common causes of short-and long-term mortality are cardiovascular and neurovascular complications [9,10,11]. The prognosis in patients with ACS is variable depending on the subtypes. Although various studies showed conflicting outcomes, a community-based study involving about 6000 cohorts found a higher 2-year mortality among patients with non-ST-elevation myocardial infarction (NSTEMI) compared with ST-elevation myocardial infarction (STEMI) after the first MI [12]. However, this has not been studied in patients with a hypertensive emergency, and most of the studies reporting on the cardiac complications of hypertensive emergencies did not provide details of the subtypes of ACS.

Based on the recommendation of the Taskforce on the universal definition of MI [13], a cardiac troponin above the 99th percentile upper reference limit with a rising and/or falling pattern and without evidence of myocardial ischemia is considered an acute myocardial injury, whereas a sustained elevation of cardiac troponin is considered a chronic myocardial injury. Myocardial injury is regarded as an entity separate from acute MI. Various studies have documented the occurrence of myocardial injury and its association with poor outcomes in various clinical conditions [14,15]. In one study, raised cardiac troponin was found to be associated with major adverse cardiac or cerebrovascular events in patients with a hypertensive crisis, independent of MI [16]. 

Notwithstanding the evidence for the prognostic significance of myocardial injury and MI in various clinical conditions [16,17], there is a paucity of data on the prevalence of myocardial injury and the subtypes of acute MI amongst patients with a hypertensive emergency. The aim of our review was to determine the prevalence of myocardial injury and MI and its subtypes in patients with hypertensive emergencies.

## 2. Materials and Methods

The protocol for this systematic review was registered with the International Prospective Register of Systematic review (PROSPERO) database (registration number CRD42022334601). The review was carried out in line with the Preferred Reporting Items for Systematic Reviews and Meta-Analysis (PRISMA) guideline [18]. The illustration in the graphical abstract was generated using BioRender.com (accessed on 17 October 2022). 

### 2.1. Search Strategy

A systematic search of PubMed, Web of Science, and EBSCOHost (MEDLINE) databases was carried out on 23 May 2022 to identify all relevant published articles that reported studies on the prevalence of hypertensive emergencies and hypertension-mediated organ damage from inception to the date of database search. The database search was restricted to studies involving adult humans. Relevant articles were searched for using the following terms: malignant hypertension, hypertensive emergency, hypertensive emergencies, hypertensive crisis, hypertensive crises, acute hypertensive crisis, acute hypertensive crises, accelerated hypertension, myocardial ischemia, unstable angina, raised cardiac troponin, raised troponin I, raised troponin T, raised cardiac enzymes, elevated cardiac troponin, elevated troponin I, elevated troponin T, elevated cardiac enzymes, acute coronary syndrome, acute myocardial infarction, ST-Elevation myocardial infarction, Non-ST elevation myocardial infarction, myocardial infarction, heart attack, and pulmonary edema. A reference list of included studies was reviewed, and additional relevant studies not captured through the database search were identified and retrieved using the Google Scholar Web search engine. A re-run of the database search was carried out on 11 September 2022 to identify relevant studies published after the initial database search. Details of the search strategy are presented in Appendix A. 

### 2.2. Criteria for Eligibility

We included hospital-based prospective, retrospective, and cross-sectional studies providing primary data on the prevalence of acute MI and/or myocardial injury in patients with a hypertensive emergency. The following were excluded from the review: studies without a clear definition of a hypertensive emergency, studies involving hypertensive disorders of pregnancy, studies involving children and adolescents, studies not published in the English language, drug-related (including recreational drug use) hypertensive emergencies, case reports (including phaeochromocytoma), editorials, letters, commentaries, reviews, unpublished conference presentations, and monographs. Studies with duplication of cohorts were reviewed and the most recent one was included. 

### 2.3. Study Selection and Data Extraction

The titles and abstracts of identified studies were screened, and the full text of included studies was retrieved and independently reviewed by MAT and EN. Any difference of opinion regarding the inclusion of studies was resolved by consensus, and reasons for exclusion after review of the full text were documented (Appendix A). The following information was extracted from the studies by MAT using a pre-defined standardized data extraction proforma: name of the first author, year of publication, country, study design, mean (SD) age of participants, the proportion of male, criteria for diagnosis of hypertensive emergency, the prevalence of hypertensive emergency, criteria for diagnosis of MI, the prevalence of MI, the prevalence of sub-types of MI, the prevalence of myocardial injury, and study period. EN cross-checked and verified the accuracy of the data extracted, and disagreements were resolved by consensus. MAT contacted the corresponding author of one of the studies [17] by email for clarification before including it in the review. 

### 2.4. Assessment of Risk of Bias

The risk of bias in the included studies was independently assessed by MAT and EN in nine areas of internal and external validity [19]. Each of the nine domains was scored 0 (poor quality) or 1 (high quality) and a composite score ranging from 0 to 9 was assigned to each study (Appendix A). The risk was categorized into low (composite score of ≥8), moderate (composite 6–7), and high (composite score of ≤5), and disagreement between the assessors was resolved by consensus. 

### 2.5. Data Synthesis and Analysis

Studies included in the review exhibited marked heterogeneity due to variation in sample size; study design; criteria for the definition of hypertensive emergency, MI, and myocardial injury; and study period, making meta-analysis untenable. As a result, we resorted to descriptive methods for data synthesis and a narrative approach for analysis. 

## 3. Results

### 3.1. Search Results 

We identified 1347 records through a database search. After removing 396 duplicates, the titles and abstracts of 951 studies were screened for eligibility, and 907 articles were excluded. The full text of 44 eligible articles was retrieved and assessed, and 14 were included. Three additional articles were identified from a reference list of the included studies. Altogether, 17 articles are included in the review. However, a study by Salvetti et al. [20] records 2 studies in one article, therefore the total number of studies analyzed was 18. A summary of the search strategy and reasons for excluding studies deemed eligible are presented in Figure 1. 

### 3.2. Study Characteristics

Characteristics of studies included in the review are presented below. There were 7545 participants with a hypertensive emergency across the final 18 studies analyzed. One of the authors reported two studies in one article and therefore, for the purpose of this review, the report was treated as two separate studies [20]. In total, 9 (50%) of the studies were carried out over one year, 3 (16.7%) over three years, and 1 (5.6%) each over ten months, sixteen months, two years, five years, six years, and twenty-seven years, respectively. Thirteen (72.2%) studies were published in the last ten years. A total of 7 (38.9%) studies were prospective, 7 (38.9%) were retrospective, and 4 (22.2%) were cross-sectional. In total, 7 (38.9%) of the studies were carried out in Europe, 4 (22.2%) in the USA, 2 (11.1%) in Africa, 2 (11.1%) in Asia, 2 (11.1%) in Brazil, and 1 (5.6%) in Mexico.

The age of the participants ranged from 46.5 (12.5) years to 76.6 (18) years. Age was not reported in one study [21]. Males predominated in 12 (66.7%) studies while females were predominant in 4 (22.2%). There were near equal proportions of males and females in two (11.1%) studies. A total of 16 (88.9%) studies reported a prevalence of hypertensive emergency ranging from 13.7% to 76.6%, 13 (72.2%) studies involved patients admitted to emergency department/emergency rooms, and 2 (11.1%) of the studies involved multiple centers. The diagnosis of hypertensive emergency was established using various guidelines in 10 (55.6%) studies, while 2 (11.1%) studies indicated using ICD codes to identify cases of hypertensive emergency. In the remaining 6 (33.3%) studies, the hypertensive emergency was diagnosed using a blood pressure of 180/110–120 mmHg in the presence of acute hypertension-mediated organ damage. 

### 3.3. Risk of Bias 

In total, 2 studies (11.1%) were categorized as having a high risk of bias [16,22], 4 (22.2%) were considered to have a low risk of bias [10,17,23,24], and 11 (61.1%) were considered as having a moderate risk of bias [9,20,21,25,26,27,28,29,30,31,32] (Appendix A). 

### 3.4. Prevalence of Myocardial Infarction

Fifteen (83.3%) studies reported the prevalence of MI ranging from 3.6% to 59.5% (Table 1). Criteria used for the diagnosis of MI included the universal definition of MI; the European Society of Cardiology 2020 guideline; and a combination of symptoms, ECG changes, and cardiac enzymes without reference to any guideline (Table 1). Of the 15 studies with information on MI, only 2 (13.3%) reported the different subtypes of MI [26,31]. One of these studies reported the prevalence of STEMI, NSTEMI, and unstable angina to be 25.2%, 19.2%, and 15.1%, respectively [31]; whereas in the other study, STEMI and NSTEMI occurred in 1.2% and 41.4%, respectively [26]. 

### 3.5. Prevalence of Myocardial Injury

The prevalence of myocardial injury was obtained in only three studies (16.7%). One study reported a prevalence of 15% using serial cardiac troponin assay [21]. In another study, 63% of the cohorts had elevated cardiac troponin without fulfilling the criteria for MI [32]. Among the Korean cohorts with a hypertensive emergency that had elevated cardiac troponin, 60.4% did not have ACS (confirmed by corresponding with the author via email) and were considered cases of myocardial injury [17].

## 4. Discussion

In this systematic review, we found: (1) a markedly varying prevalence of MI, occurring in up to 60% of patients with a hypertensive emergency; (2) a lack of information regarding the different subtypes of MI in patients with a hypertensive emergency; and (3) a paucity of reports on myocardial injury with a prevalence of up to 63% in the few available studies reporting on this. 

Close to one-third of the studies found MI in 23% to 30% of patients with a hypertensive emergency (Table 1). The highest prevalence of 59.5% was found among high-risk patients admitted to coronary care units [31], while the lowest prevalence of 3.6% was reported among cohorts in sub-Saharan Africa [28], where the prevalence of MI is generally considered to be low. However, hospitals in low- and middle-income countries may not have the capacity to comprehensively evaluate for MI and myocardial injury. Patients that presented with a hypertensive emergency who had resuscitated cardiac arrest and those that needed immediate cardiac catheterization were excluded from the cohorts in two studies, potentially resulting in the underreporting of cardiac complications [20]. A recent meta-analysis involving eight studies reported a prevalence of 18% for ACS among patients with a hypertensive emergency [7]. The high prevalence of MI in patients with a hypertensive emergency is not unexpected given that systemic hypertension is a leading risk factor for coronary artery disease and MI. In one study, coronary artery disease was present in as much as three-fourths of patients with a hypertensive emergency and elevated troponin who underwent coronary angiography [16].

The pathophysiologic mechanisms and prognosis differ in the various subtypes of ACS. However, the population of patients with a hypertensive emergency was not adequately represented in most of the studies that reported on the outcomes and prognosis in patients with ST-elevation and non-ST-elevation MI. Only two (14.7%) of the studies included in this review [26,31] provided information on the subtypes of MI, with conflicting prevalence rates for ST-elevation and non-ST-elevation MI (Table 1). Unstable angina was reported in five studies with a prevalence of up to 17.5% [16]. Although patients with unstable angina constitute a high-risk group, it is, however, debatable to what extent this can be classified as a true hypertensive emergency since an unstable angina (by definition) is not associated with features of acute hypertension-mediated cardiac damage. These uncertainties underscore the need for the standardization of the evaluation and classification of cardiac involvement in hypertensive emergencies. According to the recent European Society of Cardiology Council on hypertension position document on the management of hypertensive emergencies [33], patients presenting with acute severe, without evidence of acute hypertension-mediated, organ damage are classified as severe uncontrolled hypertension. 

Acute severe hypertension (with or without left ventricular hypertrophy) could cause myocardial oxygen supply/demand imbalance, resulting in ischemic myocardial injury and type 2 MI [13]. Endothelial dysfunction, smooth muscle cell dysfunction, and dysregulation of sympathetic innervation associated with coronary microvascular dysfunction inherent in hypertensive heart disease contributes significantly to myocardial ischemia [13]. The prevalence of myocardial injury was obtained in three studies, ranging from 15% to 63%. However, only one of the studies set out to determine the prevalence of acute myocardial injury using a serial cardiac troponin assay [21]. The paucity of reports on myocardial injury may be related to the limited application of cardiac troponin assays in patients with hypertensive emergencies. Based on the current guidelines on the evaluation and management of hypertensive emergencies, the cardiac troponin assay is recommended only when cardiac ischemia is suspected [33,34], making it difficult to determine the true prevalence of myocardial injury. The recently published international society of hypertension guideline recommends the cardiac troponin assay as an essential (minimum) investigation in the initial work-up of patients with a hypertensive emergency [35]. This will result in an increased diagnostic yield for myocardial injury and myocardial infarction, especially among patients presenting with atypical features (e.g., silent myocardial ischemia/infarction). 

The prognostic implications of myocardial injury have been established across a wide range of clinical conditions [14,15]. All-cause mortality, readmission at 30-days, and 5-year mortality rates of 11%, 21%, and 72.4%, respectively, were reported among patients with myocardial injury when compared to those without [15,36]. In one of the studies, the crude rate of major adverse cardiovascular events in myocardial injury and type 2 MI was similar to patients with type 1 myocardial infarction [15]. However, myocardial injury has not been adequately studied in patients with hypertensive emergencies and is not currently considered an acute hypertension-mediated organ injury. Given its established role in predicting adverse, cardiac, cerebrovascular, and renal outcomes [17,37], myocardial injury and indeed, other forms of subclinical target organ damage including subclinical acute kidney injury, and subtle forms of posterior reversible encephalopathy syndrome should be considered as acute hypertension-mediated organ damage in patients with a hypertensive emergency.

### Limitations

As with all systematic reviews, ours also has important limitations: (1) The studies evaluated used different criteria for the diagnosis of a hypertensive emergency and MI. This may partly be responsible for the wide variation in the reported prevalence rates. However, they mostly adopted the guideline-based definitions prevailing at the time of conducting their studies. (2) Very few studies reported on the prevalence of myocardial injury and the subtypes of MI. (3) The studies were carried out in different settings with varied populations which could have resulted in selection bias. For instance, high-risk cohorts admitted into coronary care units may have higher rates of cardiovascular complications. On the other hand, some studies excluded certain categories of high-risk groups from their cohorts, with a potential for the underestimation of the actual burden of MI. (4) Based on the current hypertension guidelines, cardiac troponin assays are not recommended in all patients with a hypertensive emergency. This could result in the under-diagnosis of both MI (especially atypical cases) and myocardial injury. (5) Only studies published in the English language were included in the review. (6) Only tertiary hospital-based studies were included in the review, resulting in the exclusion of cases of hypertensive emergency diagnosed and treated at secondary-level hospitals, especially in low- and middle-income countries. (7) We excluded case reports of secondary causes of hypertensive crisis. However, our review focused on the hypertensive emergency occurring in the patient population (primary hypertension) represented in the studies included. (8) Meta-analysis was not feasible because of the heterogeneous nature of the studies and the lack of an adequate number of studies on the myocardial injury.

## 5. Conclusions

The limitations mentioned above notwithstanding, our study has some strengths. This, to the best of our knowledge, is the first systematic review looking at the prevalence of myocardial injury and subtypes of MI in patients with hypertensive emergencies. Our findings of a varying, but overall high, prevalence of both MI and myocardial injury, albeit reported in only a relatively small number of the studies, underscores the need for more research to determine the actual prevalence of MI and its subtypes, and the prevalence of myocardial injury in patients with a hypertensive emergency. A cardiac troponin assay should be routinely carried out in all patients with hypertensive emergencies regardless of the presence of features of myocardial ischemia. Given its prognostic implications, acute myocardial injury (and other forms of subclinical target organ injury) should be classified as subclinical acute hypertension-mediated organ damage. These measures will improve the management strategies, including risk stratification, and recognition of individuals at increased risk of future cardiovascular, cerebrovascular, and renal events.

## Figures and Tables

**Figure 1 diagnostics-13-00060-f001:**
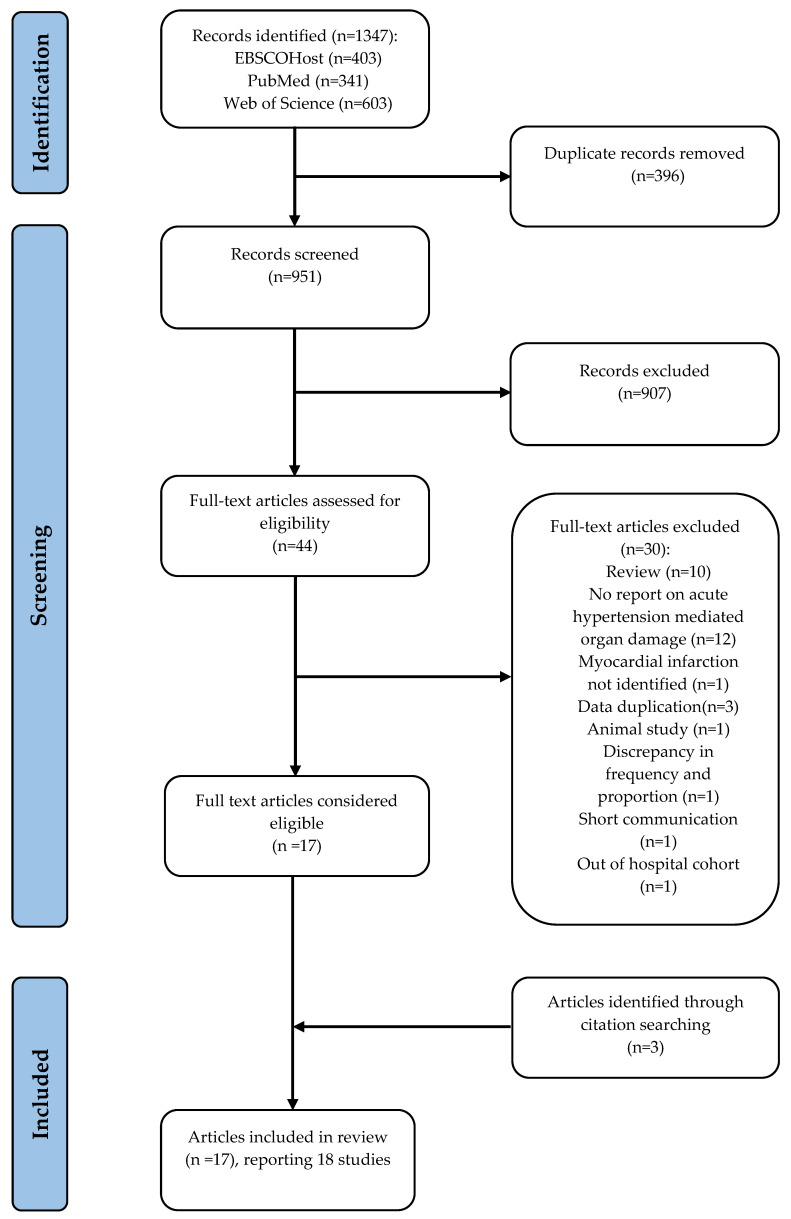
Flow diagram showing a summary of the search strategy. One article reported 2 studies, resulting in a composite of 18 studies.

**Table 1 diagnostics-13-00060-t001:** Characteristics of studies included in the review.

Author, Year, Country	Study Design	Mean Age Years (SD)	Male (%)	Diagnosis of HC	HE (%)	Diagnosis of MI	MI (%)	Sub-Types of MI/ACS (%)			NIMI (%)	Comments
								STEMI	NSTEMI	UA		
Rubin et al. [32], 2019, France	Prospective (Registry)	46.5 ± 12.5	68	180/110 aHMOD	NR	Excluded MI	NR	-	-	-	63	Cohorts comprised patients of the malignant hypertension registry from 1995 to 2017. Excluded MI. 63% had elevated cardiac troponin without signs of ongoing infarction.
Gonzalez Pacheco et al. [31], 2013, Mexico	Retrospective	62.6 ± 12.7	57.8	JNC VII	76.6	NR	59.5 (ACS)	25.2	19.2	15.1	NR	Cohorts comprised high-risk patients admitted into the coronary care unit from 2005 to 2011. Reported a high prevalence of MI and acute aortic syndrome. Provided data on subtypes of ACS.
Guiga et al. [9], 2017, France	Prospective	76.6 ± 18	46.8	ESH/ESC 2013	57.5	Symptoms ECG Troponin	13.8 (ACS)	NR	NR	NR	NR	Assessed hospital and out-of-hospital mortality in patients with hypertensive crises admitted over 12 months from January to December 2015. Short- and long-term mortality is driven by neurovascular and cardiovascular complications.
Salvetti et al. [20], 2008, Italy	Prospective	71.4 ± 14	53.9	JNC VII	20.4	3rd UD	25 (ACS)	NR	NR	NR	NR	Evaluated patients admitted over 12 months in 2008. Excluded resuscitated, cardiac arrest, and those going directly for coronary angiography.
Salvetti et al. [20], 2015, Italy	Prospective	72.5 ± 13	55.1	JNC VII	15.4	3rd UD	25 (ACS)	NR	NR	NR	NR	Evaluated patients admitted over 12 months in 2015. Excluded resuscitated, cardiac arrest, and those going directly for coronary angiography.
Pinna et al. [10], 2014, Italy	Prospective	69.9 ± 14.3	53.2	220/120 aHMOD	25.3	3rd UD	17.9	NR	NR	NR	NR	A multicenter study involving 10 Italian centers over 12 months in 2009. Reported similar rates of cardiological symptoms in hypertensive emergencies and urgencies.
Kotruchin et al. [24], 2021, Thailand	Retrospective (Registry)	65.9 ± 13.6	52.1	ACC/AHA 2017	13.7	3rd UD	6.5 (ACS)	NR	NR	NR	NR	Reviewed data of hypertension registry cohorts admitted from 2016 to 2019. Reported aHMOD involving the brain in 70% and overall in-hospital mortality of 1.6%.
Vilela-Martin et al. [30], 2011, Brazil	CS	63.4 ± 13.4	50.6	JNC VII	63.8	Symptoms ECG Troponin	25.1 (MI and UA)	NR	NR	12.1	NR	Evaluated records of patients admitted over 12 months in 2000. Found an equal proportion of UA (12.1%) and MI (13%).
Acosta et al. [21], 2020, USA	Retrospective	NR	47.6	ICD 10 Codes	NR	-	-	-	-	-	15	Evaluated myocardial injury in patients with hypertensive crisis using serial cardiac troponin assay. Found detectable cardiac troponin levels in 2/3, and myocardial injury in 15%. Excluded patients with ACS.
Zampaglione et al. [23], 1994, Italy	Prospective	67 ± 16	49.1	JNC V	24.1	NR	12 (MI or UA)	NR	NR	NR	NR	Enrolled patients with hypertensive crisis from June 1992 to May 1993. Women constituted 60% of their cohorts.
Martin et al. [29], 2004, Brazil	Retrospective	59.6 ± 14.8	55.3	JNC VI	39.6	ICD Code	13 (AMI and UA)	NR	NR	5	NR	Evaluated the medical records of patients admitted over 12 months in 2000. The most common aHMOD was cerebrovascular lesions. Reported UA separately from MI.
TajEldin M et al. [22], 2018, Sudan	CS	61.9 ± 11.8	54	JNC VII	61.7	NR	13.6 (ACS)	NR	NR	4.9	NR	Enrolled patients with hypertensive emergencies from January to October 2017. Stroke was the most common aHMOD.
Pattanshety et al. [16], 2012, USA	Retrospective	57.2 ± 16	55	ICD 10	NR	NR	30.2 (MI and UA)	NR	NR	17.5	NR	Assessed the prognostic impact of troponin on outcomes in patients with hypertensive emergencies. Found obstructive coronary artery disease in 76.5% of patients with elevated cardiac troponin. The incidence of major adverse cardiovascular or cerebral events was higher in patients with elevated troponin.
Nkoke et al. [28], 2022, Cameroon	CS	51.2 ± 16.8	50	JNC VII	58.9	NR	3.6 (ACS)	NR	NR	NR	NR	Cohorts comprised 56 patients with hypertensive emergency admitted from June 2018 to June 2019. Found acute left ventricular failure with pulmonary edema in 44.6% and ACS in 3.6%.
Kim et al. [17], 2022, Republic of Korea	Retrospective	64.6 ± 15.8	50.7	180/110	23.6	NR	NR	NR	NR	NR	60.4	In total, 60.4% of cohorts with elevated cardiac troponin did not have MI. Found higher mortality in cohorts with detectable and elevated cardiac troponin levels.
Fragoulis et al. [27], 2021, Greece	Prospective (Registry)	67.4 ± 12.9	49	180/120	27.5	NR	22.6 (ACS)	NR	NR	NR	NR	Cohorts from the registry data of the emergency department of the National referral center for percutaneous coronary intervention and heart failure. Pulmonary edema occurred in 58%, while ACS occurred in 22.6%.
Benenson et al. [26], 2018, USA	Retrospective	62.2 ± 14.87	49.7	180/120	28.3	NR	42.6	1.2	41.4	NR	NR	Cohorts predominantly African-American diabetics. Found MI in 42.6% of all participants (diabetic and nondiabetics) with NSTEMI constituting 41.4%.
Katz et al. [25], 2009, USA	CS	58 (49–70) *	51	180/110	59.4	NR	11	NR	NR	NR	NR	Studied consecutive patients with acute severe hypertension across 25 institutions in the US. In-hospital mortality is chiefly driven by intracranial hemorrhage.

ACC, American College of Cardiology; ACS, acute coronary syndrome; aHMOD, acute hypertension mediated organ damage; CS, cross-sectional; ECG, electrocardiogram; ESC, European Society of Cardiology; ESH, European Society for Hypertension; HC, hypertensive crisis; HE, hypertensive emergencies; ICD 10, International Classification of Diseases, Tenth Revision; JNC, Joint National Committee; MI, myocardial infarction; NIMI, non-ischemic myocardial injury; NSTEMI, Non-ST elevation myocardial infarction; NR, not reported; SD, standard deviation; STEMI, ST-elevation myocardial infarction; UA, unstable angina; UD, universal definition; US, the United States. * Median (interquartile range).

## Data Availability

All data relevant to the study are included in the article and uploaded as Appendix A.

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
