# Peer review of "Prevalence of Myocardial Injury and Myocardial Infarction in Patients with a Hypertensive Emergency: A Systematic Review"

_diagnostics, 2022, doi:10.3390/diagnostics13010060_

Round 1

Reviewer 1 Report

-The systematic review are good, but I'm just wondering about figure.1 it is not clear to me weather is it a table or a flowchart.

-I don't see the subtypes of the troponin (Troponin I and Troponin T) in the whole manuscript I just see in the search strategy.  

-There is no mention for myoglobin. It is one of the critical markers. 

Reviewer 2 Report

I thank the authors for the opportunity to familiarize themselves with the article «Prevalence of myocardial injury and myocardial infarction in patients with hypertensive emergency: a systematic review».  

In my opinion, the article is well written and will be of interest to the medical community.

There are a number of shortcomings, eliminating which the article will look even better:

1. It is necessary to check the introduction of abbreviations for frequently used words according to the text, such as MI and ACS, and vice versa, remove abbreviations for words rarely used - LMICs (following the guidelines for authors)

2. Add literature source â„–1

3. I cant find references to 19 literature sources in the text, this needs to be checked

4. Please, fix the design of the figure 1, at the moment it is hard to read

Reviewer 3 Report

Talle et al. conducted a systematic review on cardiac damage in hypertensive emergencies

The authors have adequately introduced the reader into theme and addressed the issue of cardiac damage in hypertensive emergency. 

I wonder why authors did not use the phrase increased troponin for search, as this formulation is much more commonly used then raised.

I believe that drug-induced emergencies were not appropriately excluded. I realize that the author applied the logic of cardiac toxicity unrelated to BP, for instance in cocaine, however, there are also many more mechanisms (most of which we do not know about) in other precipitating diseases as well. I suggest the authors to introduce these emergencies as well (as these are endorsed by contemporary guidelines), and I believe that this should not be an issue given that meta-analysis was not performed in the end (appropriately). In fact, I applaud that the authors realized the unfeasibility of performing meta-analysis due to large variation, as this is unfortunately commonly neglected in published meta-analyses. As for exclusion, unstable angina should be excluded, as it simply does not fit in the contemporary definition of myocardial damage in hypertensive emergency. In this regard, the term unstable angina is now largely being abandoned (with introduction of hs-Tn).

Figure 1. something is wrong with the styling of this figure, please revise

Discussion is overall written well, although the authors should have done better job in distinction behind mechanisms of myocardial injury in hypertensive emergencies. Namely, the authors should discuss, from pathophysiological standpoint, how STEMI may arise from hyp emergency (which is in my opinion overestimated, as this association can commonly go under "chicken or egg" conundrum).

Finally, perhaps the biggest issue of this paper is the fact that the authors published pretty much the same paper, though in from of narrative review, three months ago. As no formal analysis such as meta-analysis was performed, I see no novelty in between the two, in fact I read the first paper few months ago with great interest, and I find it better than this one. Notwithstanding, this is not mine but rather editor's choice to make.

Reviewer 4 Report

The submitted manuscript in form of a systematic review with title: "Prevalence of myocardial injury and myocardial infarction in patients with hypertensive emergency" , despite its retrospective nature of the selected observational studies coped with their respective bias and heterogeneity in the patient selection to define the prevalence of myocardial infarction in such a cohort, was clearly presented. 

The authors correctly defined the preselected criteria of myocardial infarction in such a diversative cohort according to the Universal Definition based on the the recommendation of the Task-Force. 

The presented table-overview of the relevant articles was assessed logically according to their utility in a clinical setting. 

All the potential flaws of the review were analysed and presented properly. 

However there are some minor points I would to emphasize: 

- How many of the hypertensive emergencies were at the end of secondary cause? Are there any data? 

- What about the prevalence of myocardial infarction in secondary hypertensive emergency? Was is analysed and if so how was its correlation?

- How many of the predefined myocardial infarctions with hypertensive emergencies did have takotsubo cardiomyopathy ?

Round 2

Reviewer 3 Report

The authors appropriately addressed my concerns that could have been addressed. I am nonetheless still convinced that this is a preheated version of previously published, very interesting study. I wish the authors all the best.